# Antibacterial and Anti-Fungal Biological Activities for Acrylonitrile, Acrylamide and 2-Acrylamido-2-Methylpropane Sulphonic Acid Crosslinked Terpolymers

**DOI:** 10.3390/ma13214891

**Published:** 2020-10-30

**Authors:** Reem K. Farag, Ayman M. Atta, Ahmed Labena, Salma H. AlHawari, Gehan Safwat, Ayman Diab

**Affiliations:** 1Department of Application, Egyptian Petroleum Research Institute (EPRI), Nasr City, Cairo 11727, Egypt; reem_kamal.kamel2009@yahoo.com (R.K.F.); labena.labena@googlemail.com (A.L.); 2Chemistry Department, College of Science, King Saud University, P.O. Box 2455, Riyadh 11451, Saudi Arabia; 3Faculty of Biotechnology, October University for Modern Science and Arts, 26 July Mehwar Road Intersection with Wahat Road, 6th October City 11435, Egypt; salma.hani.alhawari@hotmail.com (S.H.A.); gsafwat@msa.eun.eg (G.S.); aymanalidiab@gmail.com (A.D.)

**Keywords:** polymer, gel terpolymer, cross-linked polymer, anti-bacterial, anti-fungal

## Abstract

There is a pressing demand to synthesize polymers that have antibacterial and antifungal properties. The aim of this study was to synthesize a crosslinked hydrophilic terpolymer with acrylamide, acrylonitrile, acrylic acid, acrylamido-2-methylpropane sulphonic acid and ethylene glycol dimethacrylate as a crosslinker. The chemical structure and thermal stability of the prepared cross-linked terpolymers were confirmed by spectroscopic and thermal analyses. Moreover, the swelling experiments were performed to investigate their swelling capacity. Furthermore, the efficiency of the synthesized cross-linked polymer gels was assessed as an antimicrobial agent against Gram-positive, Gram-negative bacteria and fungal strains. The synthesized polymers showed broad inhibition effect, with more antibacterial activity by the AM4 polymer sample containing high percentage of acrylonitrile monomer in the prepared terpolymers (4 mol ratio of acrylic acid: 1 mol ratio of acrylamide: 16 mole ratio of acrylonitrile against Gram negative bacterial strain), while sample M3 terpolymer (1 mol ratio of acrylamide: 1 mole ratio acrylonitrile: 3 mole ratio of acrylamido-2-methylpropane sulphonic acid) showed a promising anti-fungal activity.

## 1. Introduction

Hydrogels are crosslinked hydrophilic polymeric three-dimensional structures that have the tendency to absorb enormous volumes of water and other biological fluids [1]. In addition, they possess a high class of water content, with physical properties as high flexibility similar to soft tissues. The process of dissolving can be done either by altering the conditions as the pH, the temperature or the ionization of the solution [2]. Crosslinking is a process in the polymer chemistry that results in a network structure depending on a multi-dimensional extension of a chain polymer by a cross-link, either ionic or covalent, which link a polymer to another [3]. The cross-linking performances of the polymers can be reversible or irreversible relying on the nature of the crosslinking [4]. The chemical method is irreversible; however, the physical and the biological are reversible by the application of stress, electricity, light, pressure, changing pH or magnetic field [5]. Conventional techniques of polymerization such as the condensation and the free-radical polymerizations were used mainly for preparation of chemically cross-linked polymers [6]. These chemical crosslinking techniques produce a degradable or a non-degradable cross-linked polymer relying on the monomer types (ionic or noni-ionic) and nature of chemical bonds [7]. Moreover, the cross-linked polymer that could be obtained from these techniques does not impose any difficulties during their application. This can be attributed to their robust cross-linking ability provided by the primary forces, whereas the cross-linked polymers that are achieved by the physical cross-linking may cause some difficulties throughout their application due to their fragile cross-linking provided by the secondary forces [8]. Bulk polymerization generates cross-linked polymers using one or more kinds of monomers without solvents and produces heterogeneous networks [9]. More than one monomer, rather than a crosslinker, was used to obtain copolymers or terpolymers when two or three monomers were used, respectively [10]. This variety of the monomers permits the production of cross-linked polymers with desired properties utilized in different applications [10]. Typically, this technique requires an addition of a small amount of a cross-linking agent during the polymer production [11]. The application of synthetic cross-linked terpolymer in life have abundantly increased through the recent years, due to their unique characteristics including mechanical strength, longer service life, the ability to absorb large amounts of water and being biocompatible [10]. The wide variety and the easily tuned properties of the cross-linked polymers were studied as a promising candidate in various fields and applications. These can be achieved via alteration of their chemical structure, concentration or preparation methods that can be suitable in many applications include drug delivery, wound dressing, contact lenses, cosmetics, tissue engineering and cardiac applications.

The cross-linked polymers work on rupturing the cell membranes of microbial cells, which further lead to leakage of the cytoplasmic content and death of the cell [12]. Several kinds of anti-microbial applications of cross-linked polymers have been developed in the recent years [13].

It was previously reported that there are several antimicrobial hydrogels based on natural and synthetic materials that are suitable for drug delivery in antimicrobial areas [14]. Chitosan cross-linked polymers have been synthesized and displayed high anti-bacterial and anti-fungal activities [15]. There are many researchers that have synthesized a cellulose-based cross-linked polymer which exhibited high mechanical strength, biocompatibility, swelling property and anti-microbial activity against *Saccharomyces cervisiae*, whereas the results displayed the possibility of the usage of the cross-linked polymers as an anti-microbial candidate [16]. Moreover, it was reported that peptide-based cross-linked polymers had displayed a tremendous anti-bacterial effect. The β-hairpin cross-linked polymer also possessed an anti-microbial effect [17]. Although the anti-microbial cross-linked polymers have demonstrated a remarkable activity against microbes, it was found out that the interaction between the polymer and the cell membrane was nonspecific, thus causing, in most cases, death of the mammalian cells [18]. The biopolymers were widely used due to their natural availability beside their bioactivity. The synthetic crosslinked polymers require attention to select the desired monomers having lower toxicity and higher biological activity. Recent works proved that the combination of synthetic anti-microbial polymers and antibiotics was used to evade problems of drug resistance [14]. In this research, new crosslinked polymers were prepared in order to utilize the cross-linked polymer as an anti-microbial agent to decrease the toxicity associated [19]. The synthetic antimicrobial hydrogel based on polyvinyl alcohol and polyvinyl-pyrrolidone having excellent swelling capacities was applied in wound therapy with great ability to enhance epithelialization and reduce loss of skin grafts [20]. There are thermoresponsive hydrogels based on N-isopropylacrylamide that exhibited strong anti-microbial activity, which, besides their high biocompatibility with cells, were used for biomedical application [14,21]. It was previously reported that the anti-microbial crosslinked copolymeric hydrogels based on acrylamides were prepared by radical crosslinking polymerization technique [21,22]. In this respect, the investigation of anti-microbial and anti-fungi activities of new hydrogels rather than crosslinked homopolymers or copolymers will offer antimicrobial hydrogels that can be used for drug delivery in antimicrobial areas. In the present work, different terpolymers were synthesized based on monomers, acrylonitrile (AN), acrylic acid (AA), acrylamide (Am), 2-acrylamido-2-methylpropane sulfonic acid (AMPS), benzoyl peroxide (BP) as an initiator and ethylene glycol dimethacrylate (EGDMA) cross-linker. These monomers were selected on the basis of their reactivity ratios to obtain neutral and ionic terpolymers. Moreover, EGDMA was selected as a reactive crosslinker to control the crosslinking densities of the prepared terpolymers. The objective of this study was to test the efficacy of the synthesized cross-linked terpolymer gels as anti-bacterial agents against Gram-positive and Gram-negative bacteria in addition to their anti-fungal activity.

## 2. Experimental

### 2.1. Materials

Acrylonitrile, acrylic acid, 2-acrylamido-2-methylpropane sulfonic acid (AMPS), benozyl peroxide, acrylamide, ethylene glycol dimethacrylate crosslinker and ethanol were received from Sigma-Aldrich Chemicals Co (Missouri, MO, USA). Trypticase soy broth (TSB) or trypticase soy agar (TSA) (Difco Co; Becton Dickinson, Sparks, MD, USA) were used for cultivation for bacterial strains. Sabouraud dextrose broth (SDB) or Sabouraud dextrose agar (SDA) (Difco, Sparks, MD, USA) were used for fungal strain cultivation.

### 2.2. Preparation of Cross-Linked Polymers

Nine terpolymers with different concentration ratios were synthesized, and their constituents are shown in Table 1. The sample defined as (AM0) was prepared by the addition of AA (80 mol %; 0.72 g) and AMPS (20 mol %; 0.515 g) in the presence of EGDMA (1 Wt.% related to monomers weight), BP (0.1089 g) and distilled water (50 mL). The sample (AM1) was prepared as AM0 in the presence of AN (0.53g) as the third monomer (mole ratio of AA: AMPS:AN is 4:1:4). The samples AM2, AM3 and AM4 were prepared using the same ingredients as (AM1) but with different concentrations of AN, which were 1.06, 1.59 and 2.12 g, respectively. AM1, AM2, AM3 and AM4 moles ratios of AA: AMPS:AN are 4:1:4, 4:1:8, 4:1:12 and 4:1:16, respectively. The second batch of the samples was consisted of five samples (M0, M1, M2, M3 and M4). The sample (M0) was prepared by the addition of Am (50 mol %; 0.78 g), AN (50 mol %; 0.53 g), EGDMA (1 Wt.% related to monomers weight), BP (0.1089 g) and distilled water (50 mL). The sample (M1) was prepared using AMPs (0.515 g), Am (0.78 g) and AN (0.53 g) and the same concentrations of BP, EGDMA solubilized into distilled water (50 mL). The mole ratio of AMPS: Am:AN: is 1:1:1. The samples M2, M3 and M4 were prepared using the same ingredients that were used in M1 but with different concentrations of AMPS, Am and AN, which were 2:1:1, 3:1:1 and 4:1:1 Wt.%, respectively. All solutions were stirred for 15 min until the solutions became clear under N_2_ atmosphere and sealed in test tubes. The solutions were placed in an oven at 85 °C for 4 to 5 h. After obtaining the cross-linked terpolymers were rinsed in distilled water in order to remove any unreacted monomers and were then followed by filtration and drying of the gel in vacuum oven and storing them for later evaluations.

### 2.3. Characterization by FTIR (Fourier Transform Infrared) Spectroscopy

The Fourier Transform Infrared Spectroscopy (FTIR; PerkinElmer 2000 FT-IR spectrometer, Waltham, MA, USA) was used to characterize the structural arrangement and the functional groups of the cross-linked polymer. The samples were prepared in a pellet form and diluted using KBr with 1/200 (w/w) of samples/KBr. The thermal stability of the crosslinked polymers was obtained using thermogravimetric analysis (TGA; NETZSCH STA 449 C instrument, New Castle, DE, USA) with a temperature rate of 10 °C/min, under dynamic flow of nitrogen 20 mL/min.

### 2.4. Swelling Property Measurement

The cross-linked polymers were sliced to 2 mm thickness and 3 mm diameter. Furthermore, the dried gels were left to swell in distilled water at temperature of 25 °C for one day to achieve the equilibrium swelling. The swollen cross-linked polymer gels were removed from the water after 5 min, 10 min and 20 min up to 110 min; then, the gels were dried using filter paper, weighed and placed back in the water. The measurement of the degree of swelling continued until a constant weight was achieved for each sample. The swelling degree in g/g of each gel was calculated as the following relation:(1)$$\mathrm{Degree}\;\mathrm{of}\;\mathrm{swelling}\;g/g\;=\;\frac{\left(\mathrm{Wet}\;\mathrm{weight}-\mathrm{Dry}\;\mathrm{weight}\right)}{\mathrm{Dry}\;\mathrm{weight}}$$

### 2.5. Application of the Synthesized Cross-Linked Terpolymers as Anti-Microbial Agents

#### 2.5.1. Test Organisms

Gram-positive bacteria: *Bacillus subtilis* (ATCC 6633), Gram-negative bacteria: *Escherichia coli* (ATCC 8739) and *Candida albicans* (ATCC 10231) were obtained from the American Type Culture (ATCC; Rockville, MD, USA).

#### 2.5.2. Cultivation Conditions and Anti-Microbial Activity

The anti-microbial activity test of the superabsorbent cross-linked terpolymers were carried out under aseptic techniques using the modified agar well diffusion method. The Gram-positive and the Gram-negative bacterial strains, *Bacillus subtilis* (ATCC^®®^ 6633) and *Escherichia coli* (ATCC^®®^ 8739), were streaked on trypticase soy agar (TSA) plates using a loop wire, while the fungal strain *Candida albicans* (ATCC^®®^ 10231) was streaked on a sabouraud dextrose agar (SDA) plate. Then, a sterile 10 mm Cork borer was used in order cut two well pores in each agar plate. Furthermore, the cross-linked terpolymers (100 µL of M3 and AM4) were introduced into the cut wells that were inoculated with bacterial and the fungal strains. Then, the plates were left to incubate for overnight at 37 °C and for 48 h at 35 °C regarding the bacterial strains and the fungal strain, respectively. Afterwards, the anti-microbial activity was assessed by measuring the diameter (mm) of the inhibition zone formed around the cross-linked polymers in three different fixed directions. The growth pattern of the microbes in the presence or absence of the polymers was compared using sterile water as a negative control, and standard antibiotics. Tetracycline (TE) (100 ppm), amoxicillin (AMC) (100 ppm) and Fluconazole (Flu) (100 ppm) were used as positive controls. Duplicates were sustained for each sample and the inhibition zone measurement was repeated twice.

#### 2.5.3. Minimum Inhibitory Concentration (MIC) and Minimum Bactericidal/Fungicidal (MBC/MFC) Concentrations

“The minimum inhibitory (MIC) concentrations” [23] of the cross-linked terpolymers (M3 and AM4) were estimated using a two-fold micro dilution method in 96-well micro-titer plates with modifications [24]. The bacterial and fungal strains-inocula were prepared as the Clinical Laboratory Standards Institute (CLSI) method reported [25,26]. The bacterial inocula were 1–2 × 10^8^ CFU/mL for Gram-positive bacteria (*Bacillus subtilis*) and 1–2 × 10^9^ CFU/mL for Gram-negative bacteria (*Escherichia coli*) and fungal inoculum was composed of 5 × 10^6^ CFU/mL. In total, 100 µL of the cross-linked terpolymers (M3 and AM4) were serially diluted onto the micro-titer plates using trypticase soy broth (TSB) and sabouraud dextrose broth (SDB) for the bacterial strains and the fungal strain, respectively, and further inoculated with 100 µL of the microbial inocula parallel with a positive-control (inoculated without the cross-linked terpolymers (M3 and AM4)) and a negative-control (only sterile media). Afterwards, the micro-titer plates were incubated under aerobic conditions for an incubation period of 20 h at 37 °C and 48 h at 35 °C for the bacteria and the fungal strains, respectively. Following [27], “the MIC was determined as the lowest concentration of the cross-linked terpolymers (M3 and AM4) that inhibits the development of visible bacterial, and fungal growth on cultivated media after an incubation period.”

In order to estimate “the minimum bactericidal/fungicidal concentrations (MBC/MFC)” of the cross-linked terpolymers (M3 and AM4) “needed to indicate 99.5% killing of the original inoculum,” 10 µL was taken from the wells with no observed growth and further sub-cultured onto their agar media (TSA and SDA for bacterial and fungal strains, respectively) [28].

## 3. Results and Discussion

The reactivity ratios of AA (M1) with AN (M2) are r1 = 1.188 and r1 = 0.057 form block copolymer in the presence of K_2_S_2_O_8_ as initiator in aqueous solution that reduced the reactivity of AN [28]. The reactivity ratio of AN (M2) with AMPS (M3) are r2 = 0.193 and r3 = 0.162 indicates that the reactivity of AN towards copolymerization was increased by the incorporation of AMPS as well as AMPS/AA. Therefore, at all the compositions of the initial monomer mixture the copolymers are enriched in AA units. The radical polymerization of Am with AN and AMPS forms random terpolymers with uniform distribution of AMPS and AN unit along the macromolecular chain to reduce the hydrodynamic resistance to aqueous flow that increased its usefulness as a drag reducing additive to extract the oil [28]. The samples AM0, AM1, AM2, AM3 and AM4 were designed to stabilize the moles contents of AMPS and AA and increases AN contents to increase the nitrogen contents in the crosslinked polymers that enhanced by the crosslinking polymerization of AMPS/AN. It was previously reported that the presence of quaternary nitrogen and crosslinking using EGDMA will increase the mechanical stability of the crosslinked polymers beside increasing biocidal activities [29]. The crosslinking of materials improves their easy separation, recovery and antimicrobial activity [30]. Moreover, antimicrobial polymers can destroy the bacterial membrane that may help to prevent antibiotic resistance [31]. The samples of M0, M1, M2, M3 and M4 also were designed on the basis of increasing nitrogen contents with increasing the Am and AN contents in the terpolymers AMPs/Am/AN. In this respect, the crosslinking scheme of AA/AMPS/AN or AMPS/Am/AN in the presence of EGDMA as crosslinker were represented in Scheme 1.

### 3.1. Characterization of the Crosslinked Terpolymers

The FTIR analysis was used to illustrate the structural and the functional groups of the cross-linked polymer samples based on AA/AMPS/AN (AM) and AMPS/Am/AA (M). In this respect, FTIR spectra of AM1, AM4 and AM2 were selected and represented in Figure 1a–c to illustrate the effect of increasing AN contents in the monomer feedstock on the chemical structures of the cross-linked terpolymers. Figure 2a–d represents the effect of increasing Am/AN contents on the chemical structures of crosslinked M terpolymers. The intensity CN stretching vibration bands at 2240 cm^−1^ were used to confirm that the AM4 (Figure 1c) has higher AN contents more than that obtained for AM1 (Figure 1a) and AM2 (Figure 1c). The broad band ranged from 3435 to 3600 ${\mathrm{cm}}^{-1}$ demonstrated the stretching vibration of –NH and COOH was appeared in all AM spectra to confirm the incorporation of AA and AMPS in terpolymer structures (Scheme 1 and Figure 1a–c). The disappearance of =CH stretching vibration and appearance of aliphatic CH stretching vibration at 2926 and 2855 ${\mathrm{cm}}^{-1}$ elucidate the crosslinking radical polymerization of AA/AMPS/AN (Figure 1a–c). The band at 1750 ${\mathrm{cm}}^{-1}$ assigned for the stretching vibration of the C=O ester bond confirms the incorporation of EGDMA crosslinker in the chemical structures of terpolymers (Figure 1a–c). Moreover, the band at 1300 ${\mathrm{cm}}^{-1}$ and 650 ${\mathrm{cm}}^{-1}$ assigned the asymmetric stretching of S=O and S-C bonds, respectively, elucidates the incorporation of AMPS in terpolymers (Figure 1a–c).

The FTIR spectra of M polymer (Figure 2a–d) elucidate that the incorporation of AMPS in the polymers M4 (Figure 2a), M2 (Figure 2b) and M3 (Figure 2c) increases the intensity of CN stretching vibration by 2312–2350 ${\mathrm{cm}}^{-1}$ compared to M0 (Figure 2d, AN/Am copolymer) as designed. The characteristic bands related to AMPS and AN represented in the previous section of AM samples were appeared in spectra of AMPS/Am/AN (Figure 2a–c). The C=O stretching vibration band of Am and AMPS appeared at 1645–1668 cm^-1^ in M polymer spectra (Figure 2a–d).

The thermal stability data of the M and AM polymers were determined from their TGA thermograms and represented in Figure 3a,b, respectively. It was previously reported that the resistance of AMPS/AN/Am linear terpolymer to salt and temperature were affected by the terpolymer compositions [32]. It was observed that the heating of linear AMPS/Am/Am terpolymer produced crosslinked polymers due to cyclization of amide groups with the formation of six membered imide rings [32]. It was also found that the lowering of AMPS and AN moles contents relative to Am increases the terpolymer resistance to thermal degradation. Moreover, increasing of AMPs content more than AN improve the terpolymer thermal stability and resistance to hydrolysis under hydrothermal conditions due to manifestation of the electrostatic effect [32]. These results agree with our recorded data for M samples. It was noticed that the cross-linked AN/AM copolymer (M0) and AMPS/Am/AN terpolymer (M4) did not loss any weight at temperature below 120 °C rather than M1–M3 (Figure 3a) that lost approximately 10 Wt.% from their original weights. This Wt.% was related to humidity adsorbed by gels that reduced with the absence of AMPS (M0) and increasing Am/AN ratios (M4). The AMPS contains amide and sulfonic groups that increase their bonding with water humidity. Moreover, it was also noticed that the initial degradation temperatures (IDT) of terpolymers were increased with increasing of Am/AN contents and lowering AMPS contents. IDTs of M1, M2, M3 and M4 were recorded as 245, 320, 285 and 350 °C, respectively. The increasing of AMPs content in M1 reduces the thermal stability of crosslinked Am/AN copolymer (M0) that was degraded at IDT of 280 °C (Figure 3a). The sulfonic group of AMPS increases the polymer degradation. The increasing of Am/AN contents increases the remained residual percentage (RS %) above 650 °C for M4 > M3 > M2 > M1 > M0. The RS % values above 650 °C of M4, M3, M2, M1 and M0 are 35, 26, 22, 12 and 2 Wt.%, respectively. The RS % were referred due to intra- and intermolecular condensation of amide groups of the AMPS and Am at elevated temperatures with the formation of six-membered imide rings, which results in their cross-linking through intermolecular cyclization with the formation of a three-dimensional structure. This observation was proved from increasing of the RS % of M samples with increasing Am/AN contents (Figure 3a). Moreover, the RS % of AM samples (Figure 3b) were decreased than that recorded with M samples due to replacement of Am with AA. The presence of AA in the AM samples reduces both IDT and RS % of AM samples due to decarboxylation of COOH groups that increases the thermal degradation of terpolymers without formation of crosslinked rings such as that obtained with M samples.

### 3.2. Swelling Analysis Measurements

It is very important to investigate the swelling and water uptake of the crosslinked polymer that can be used for biocide or antimicrobial biofilms to investigate the effect of water uptake on the gel shapes, and mechanical stability without the addition of biocidal species [31]. In this respect, the photos of the swelled gels were represented in Figure 4 to confirm the mechanical and shape stability of M and AM samples. The results of the swelling capacities or water absorbance versus times for all prepared samples were represented in Figure 5a,b. It was noticed that the swelling of the cross-linked terpolymers M1–M4 and AM1–AM4 were increased more than that obtained for AA/AMPS (AM0 and AN/Am (M0)) cross-linked copolymers (Figure 5a,b). It was also demonstrated that the swelling ability of the cross-linked terpolymer samples was increased by increasing the concentration of the acrylamide and the acrylonitrile without release or solubilization of the cross-linked polymer to confirm the crosslinking of terpolymers. The highest swelling was achieved for M samples (Figure 5a) at 85.7 for the sample M4 which occurred when the concentrations of the total weight of AMPS, Am and AN were increased to 2.06, 3.12 and 2.12 g, respectively. In the second batch of the samples (Figure 5b; AM0, AM1, AM2, AM3 and AM4), the cross-linked polymer samples were prepared with same ingredients with different AN contents, where the highest swelling was achieved in the sample AM4 of 97.33 due to the increasing of the AN weight contents increased from 1.59 to 2.12 g. The swelling was tested for the cross-linked polymer samples with different ratios of acrylamide, the result of the swelling was 63.8 for the acrylamide ratio of 50%; however, upon increasing of the acrylamide content ratio from 50% to 66%, the swelling achieved was 123.57, which was then decreased to 78.550 over time. Thus, this proves that the swelling behavior can be controlled by changing the content of the monomers. However, it was reported that the swelling behavior of the cross-linked polymers that were made based on the free radical polymerization was different due to varying ratios of acrylamide and acrylonitrile (Am/AN) [33]. The swelling was decreased with the increasing the cross-linking ratio due to the acrylonitrile (AN) concentrations, while the polymer that was composed of higher acrylamide Am content has shown a higher swelling property in comparison to the polymer that has a higher content of acrylic acid (AA), which was inconsistent with the present results. Furthermore, a study was done using cross-linked polymers with different ratios of acrylamide, whereas the ratios for the Am were 40%, 50%, 60% and 80%. The experimental result regarding the effect of the Am on swelling was found to be that the swelling of the gel increased from 3.10 to 3.50 when the Am content ratio was increased from 40% to 50%, and when further increased from 50% to 80%, the ratio of the swelling considerably increased from 5.46 reaching 14.10. Thus, this indicates that the swelling ratio increases with the substantial increasing of the acrylamide ratio, attributed to the increased proportion that allows more effective binding sites to be available in the polymer chain [34]. It can also be concluded that the increasing of the water absorbance increased for terpolymers due to lowering of the crosslinking densities and higher reactivity of monomers towards terpolymerization rather than polymerization with EGDMA crosslinker.

### 3.3. Antimicrobial Activity

The result represents the agar well diffusion plates that were inoculated with the test micro-organisms (*Bacillus subtilis*, *Escherichia coli* and *Candida albicans*) followed by the introduction of the cross-linked terpolymer samples (M3 and AM4) where (a) represents the M3 polymer sample and (b) represents the AM4 sample. AM4 was selected due to its higher swelling in water (Figure 6b) besides the higher AMPS and AA contents that facilitate binding of polymers with anionic bacterial membrane due to electrostatic interactions. M3 was also selected due to the moderate molar ratio of AMPS to AN and Am monomers and moderate swelling characteristics (Figure 6a). The anti-microbial activity of the superabsorbent cross-linked terpolymers (M3 and AM4) against the Gram-positive bacterial strain (*Bacillus subtilis*), Gram-negative bacterial strain (*Escherichia coli*) and the fungal strain (*Candida albicans*) is stated in Table 2. The synthesized cross-linked polymers displayed a broad range of anti-bacterial and anti-Candida activity with the clearing inhibition zones ranging from 22–35 mm compared to the standard antimicrobial agents. Moreover, the cross-linked terpolymers (M3 and AM4) represented a higher anti-bacterial activity against the Gram-positive (25.5–35 mm) than for the Gram-negative bacteria (22–25.5 mm). This result might be in attribution to the cell wall differences between the Gram-negative and the Gram-positive bacteria [35]. As the cell wall of the Gram-positive bacteria is fully composed of peptide poly-glycogen, which is made up of many pores that allow the diffusion of the foreign molecules into the cell without strain [36]. The M3 cross-linked terpolymer has demonstrated the lowest anti-bacterial activity, while the AM4 cross-linked terpolymer has displayed the highest anti-bacterial. The cross-linked polymer AM4 result regarding the fungal strain (*Candida albicans*) was 32.5 mm, while the cross-linked polymer compound M3 showed activity of 33 mm. Furthermore, this proved that all the cross-linked terpolymers had demonstrated anti-bacterial and anti-fungal activity, with the highest anti-candida activity being the M3 cross-linked polymer and the highest for the anti-bacterial activity being the AM4 cross-linked polymer compound. The minimum inhibitory concentration (MICs), minimum bactericidal concentration (MBCs) and minimum fungicidal concentration (MFCs) of the superabsorbent cross-linked terpolymers (M3 and AM4) were listed in Table 3. The AM4 cross-linked polymer has exhibited the lowest MIC/MBC of (15.62/62.5 ppm), (15.62/62.5 ppm) and (15.6/31.2 ppm) compared to M3 cross-linked polymer compound of (31.25/62.5 ppm), (31.25/62.5 ppm) and (15.6/31.2 ppm) against *Bacillus subtilis*, *Escherichia coli* and *Candida albicans*, respectively. This result was attributed to the fact that increases content of AA and AMPS moiety, which leads to reduced pH value [37]. The reduction in the pH values is a well-known phenomenon to cause stress on the cells, discrupting the cells’ homeostasis [38]. In addition, sustained exposure to increased acid concentrations would result in cell membrane destruction [39]. Consequently, it renders the cell to being attacked by small fatty acids, which can eradicate the bacteria relying on the concentration of the acids and the value of the pH [38]. Moreover, it has been reported that some acids are identified to be an anti-fungal and anti-viral agent, such as ferulic acid, which has strong anti-fungal activity [40]. Furthermore, it is interesting to note that the polymeric sample (AM4) exhibited high anti-bacterial activity, which agrees with the study of [41]. In addition, it has been shown that the inhibitory effect of the cross-linked terpolymer samples were increased with the increasing the amounts of both the acrylic acid and the acrylonitrile monomers [39]. In addition, it may also be due to the swelling ability of the cross-linked polymers, which enhances the diffusion of the active monomers inside the pathogen, which leads to an enzyme disturbance, which is responsible for growth, thus destroying the pathogenic microorganisms. Whereas with increasing acid concentrations, the swelling ability of the cross-linked polymer increases, thus improving contact surface of the microorganism and the polymer samples [42]. It was also reported that the binding of the chitosan with DNA acquired from a microorganism was eventually result in inhibition of the mRNA and protein synthesis through penetrating the nucleus of the microorganism by the activity of the chitosan [43,44].

## 4. Conclusions

Based on the reported results in this study, it was concluded that the cross-linked hydrophilic terpolymer gel samples were promising candidates to be used in the anti-bacterial and anti-fungal applications. The polymeric sample (AM4) has established the highest anti Gram-positive bacterial activity ranging from 25.5 to 35 mm, while the M3 gel sample has displayed the highest anti-fungal activity result of 33 mm. Thus, this proves that the cross-linked polymers are seen as a powerful tool for various medical applications, such as wound dressings, urinary tract coatings, contact lenses, treatment of osteomyelitis, catheter-associated infections, gastrointestinal infections and so on.

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
