# Peer review of "Antibacterial and Anti-Fungal Biological Activities for Acrylonitrile, Acrylamide and 2-Acrylamido-2-Methylpropane Sulphonic Acid Crosslinked Terpolymers"

_materials, 2020, doi:10.3390/ma13214891_

Round 1

Reviewer 1 Report

The manuscript deals with a very interesting work, but in its present form it is not very clear to readers. First of all, the English language/grammar should be greatly improved, as there are several sentences that seem incomplete, wrong or unclear (for example, the first sentences in paragraph 2.5.4). I recommend having a native English speaker reviewing the manuscript. Then, in addition to the language, I also have the following suggestions/comments:

  • Abstract: samples AM4 and M3 are cited at the end of the abstract, but readers of the abstract have no idea of what these samples are. Please briefly state what they are, so that the abstract is clear.
  • Introduction: you use several times the word ‘anti-biological’, but that does not sound correct. Maybe anti-microbial? Anti-bacterial?
  • Methods: although you describe the preparation of your different samples in the text, it is not easy to remember what the various AM0, etc. stand for, while reading the manuscript. As such, it would be useful to add a synoptic Table for the sample synthesis, to remind what each sample is.
  • Methods/Results: in the results, you describe TGA curves, but this section is now missing in the methods. Please add it.
  • Methods/Results: in the methods, you reported some time points for the swelling measurements (from 5 to 30 min, and then 1 day, etc.), but in Figure 5 the time points seem different (up to 120 min). I guess the methods should be reviewed. Moreover, in Fig. 5 it seems that swelling ratios (g/g) are reported instead of the percentage indicated in the equation of the degree of swelling at page 3. Please adjust the equation.
  • Methods: it is not clear whether the sections Bacterial inoculum and Fungal inoculum are two different paragraphs, or parts of paragraph 2.5.4. Actually, they seem to make more sense if included in paragraph 2.5.3. Please review/adjust this part.
  • Results (terpolymer synthesis): please define the reactivity ratios or how they are calculated.
  • Results (FTIR): lines in Fig. 1 are too thick. Moreover, it would be useful to highlight the peaks of interest, as done in Fig. 2.
  • Results (biological activity): it is not clear why only samples M3 and AM4 were used for these tests. I suggest to explain this selection in the text.

Author Response

Response to reviewers

First of all, I really appreciated the constructive criticisms of the reviewers.  I have addressed each of their concerns as outlined below.

Reviewer #1:

  • English language/grammar should be greatly improved, as there are several sentences that seem incomplete, wrong or unclear (for example, the first sentences in paragraph 2.5.4). I recommend having a native English speaker reviewing the manuscript. Then, in addition to the language, I also have the following suggestions/comments:

"Author reply"

Thanks a lot for your valuable comments, English language has been revised.

  • Abstract: samples AM4 and M3 are cited at the end of the abstract, but readers of the abstract have no idea of what these samples are. Please briefly state what they are, so that the abstract is clear.

"Author reply"

Done.

  • Introduction: you use several times the word ‘anti-biological’, but that does not sound correct. Maybe anti-microbial? Anti-bacterial?

"Author reply"

Thanks for your notice, corrected.

  • Methods: although you describe the preparation of your different samples in the text, it is not easy to remember what the various AM0, etc. stand for, while reading the manuscript. As such, it would be useful to add a synoptic Table for the sample synthesis, to remind what each sample is.

"Author reply"

Table 1 is inserted in text.

  • Methods/Results: in the results, you describe TGA curves, but this section is now missing in the methods. Please add it.

"Author reply"

Done in the characterization section

Methods/Results: in the methods, you reported some time points for the swelling measurements (from 5 to 30 min, and then 1 day, etc.), but in Figure 5 the time points seem different (up to 120 min). I guess the methods should be reviewed. Moreover, in Fig. 5 it seems that swelling ratios (g/g) are reported instead of the percentage indicated in the equation of the degree of swelling at page 3. Please adjust the equation.

"Author reply"

Done.

  • Methods: it is not clear whether the sections Bacterial inoculum and Fungal inoculum are two different paragraphs, or parts of paragraph 2.5.4. Actually, they seem to make more sense if included in paragraph 2.5.3. Please review/adjust this part.

"Author reply"

Thanks for your notice, corrected.

  • Results (terpolymer synthesis): please define the reactivity ratios or how they are calculated.

"Author reply" reactivity ratios were not determined here because it is not possible determined for the crosslinked polymers. Reactivity ratios of copolymers reported here from the literature and they were obtained from copolymerization of monomers at low conversion approx10%.and analyzed the copolymer compositions by spectroscopic or elemental analyses. The different between monomers molar ratios in the monomers and copolymers composition were calculated and determined from equation of kelen-tudos or finmanross methods.

  • Results (FTIR): lines in Fig. 1 are too thick. Moreover, it would be useful to highlight the peaks of interest, as done in Fig. 2.

"Author reply"

  • They were marked
  • Results (biological activity): it is not clear why only samples M3 and AM4 were used for these tests. I suggest explaining this selection in the text.

"Author reply"

M3 and AM4 are selected because they have the highest swelling capacity. Moreover the swelling ability of the cross-linked polymers increasing their efficiency as Antibacterial and Anti-fungal Biological Activities, which works on enhancing the diffusion of the active monomers inside the pathogen inducing enzyme disturbance of which are responsible for growth thus destroying the pathogenic microorganism, whereas with increasing acid concentrations, the swelling ability of the cross-linked polymer increases, thus improving contact surface of the microorganism and the polymer samples (Malmsten, 2011). Ref :Malmsten, M. (2011). Antimicrobial and antiviral hydrogels. Soft Matter7(19), 8725-8736.

Reviewer 2 Report

Comments on the Manuscript “Antibacterial and Anti-fungal Biological Activities

for Acrylonitrile, Acrylamide and 2-Acrylamido-2- Methylpropane Sulphonic Acid Crosslinked Terpolymers” referenced by materials-969320

The paper reports on the antibacterial and anti-fungal properties of some synthetic-based cross-linked terpolymers. I consider that the manuscript is not suitable in the present form for publication in Materials journal because this paper has serious drawbacks, as it is presented in detail below:

  1. English needs improvement through out the manuscript and spelling should be checked thoroughly. There are many spelling mistakes and some phrases are not clear in the original paper. For example, in the introduction “Conventional techniques of polymerization such as the condensation and the free-radical polymerizations.”; “They are used mainly for the chemically cross-linked polymers”; “This variety of the kinds of monomers permits…” are not clear and should be rewritten.

  1. The application of synthetic cross-linked terpolymer in life have abundantly increased through the recent years, due to their unique characteristics including mechanical strength, longer service life, the ability to absorb large amounts of water and being biocompatible” should be sustained by references.

  1. The cross-linked terpolymers reported in this paper are based on synthetic polymers; therefore the discussion on natural polymers included in the introduction part is not related to the current study and should be removed. I recommend the addition of a similar discussion on for synthetic polymers already reported in literature.

  1. Lines 60-61: “anti-biological applications of cross-linked polymers” need clarifications and references.

  1. In the Introduction, the novelty of the work is weakly emphasized. The synthesis of the acrylonitrile, acrylamide and 2-Acrylamido-2- methylpropane sulphonic acid cross-linked terpolymers has been already reported, refs. 20-22. In this context, the authors’ motivation for selecting this system needs stronger explanations. Why this system is better than others? The

  1. The Results and discussion section should be revised because some parts of it are ambiguous, difficult to follow, and don’t emphasize the value of this work. A clear, fluent and more critical interpretation of the results regarding the antibacterial and anti-fungal properties should be provided.

  1. In the FTIR spectra, Figure 1, please label the discussed peaks included in the main text. The optical pictures, Figure 4, have very low resolution and do not show any swelled gels.

  1. In Table 1 and Table 2, please correct the name of the fungal bacterium as “Candida” and not “Candia”.

  1. Where these cross-linked terpolymers are going to be used in real life? What is the relationship between antifungal properties and oil recovery? The last sentence included in the Conclusion part is not supported by the results presented in this paper.

Based on the comments given above, my recommendation is “Major Revision”.

Author Response

Reviewer 2:

detail below:

  • English needs improvement throughout the manuscript and spelling should be checked thoroughly. There are many spelling mistakes and some phrases are not clear in the original paper. For example, in the introduction “Conventional techniques of polymerization such as the condensation and the free-radical polymerizations.”; “They are used mainly for the chemically cross-linked polymers”; “This variety of the kinds of monomers permits…” are not clear and should be rewritten.

 "Author reply"

Done

 “The application of synthetic cross-linked terpolymer in life have abundantly increased through the recent years, due to their unique characteristics including mechanical strength, longer service life, the ability to absorb large amounts of water and being biocompatible” should be sustained by references.

 "Author reply"

Done

The cross-linked terpolymers reported in this paper are based on synthetic polymers; therefore the discussion on natural polymers included in the introduction part is not related to the current study and should be removed. I recommend the addition of a similar discussion on for synthetic polymers already reported in literature.

 "Author reply"

This part discuss literature survey on antimicrobial activity of synthetic and natural polymers

  • Lines 60-61: “anti-biological applications of cross-linked polymers” need clarifications and references.

"Author reply"

Done.

  • In the Introduction, the novelty of the work is weakly emphasized. The synthesis of the acrylonitrile, acrylamide and 2-Acrylamido-2- methylpropane sulphonic acid cross-linked terpolymers has been already reported, refs. 20-22. In this context, the authors’ motivation for selecting this system needs stronger explanations. Why this system is better than others? The

 "Author reply"

  • It was previously reported that the antimicrobial crosslinked copolymeric hydrogels were prepared by radical crosslinking polymerization technique [20-22]. In the present work, different terpolymers were synthesized based on monomers, acrylonitrile (AN), acrylic acid (AA), acrylamide (Am), AMPS, benzoyl peroxide (BP) as an initiator and ethylene glycol dimethacrylate (EGDMA) cross-linker. These monomers were selected on the basis of their reactivity ratios to obtain neutral, and ionic terpolymers. Moreover, EGDMA was selected as reactive crosslinker to control the crosslinking densities of the prepared terpolymers.
  • The Results and discussion section should be revised because some parts of it are ambiguous, difficult to follow, and don’t emphasize the value of this work. A clear, fluent and more critical interpretation of the results regarding the antibacterial and anti-fungal properties should be provided.

"Author reply"

Done.

  • In the FTIR spectra, Figure 1, please label the discussed peaks included in the main text. The optical pictures, Figure 4, have very low resolution and do not show any swelled gels.

 "Author reply"

The peaks were marked

  • In Table 1 and Table 2, please correct the name of the fungal bacterium as “Candida” and not “Candia”.

 "Author reply"

Corrected.

  • Where these cross-linked terpolymers are going to be used in real life? What is the relationship between antifungal properties and oil recovery? The last sentence included in the Conclusion part is not supported by the results presented in this paper.

Author reply"

Corrected, sorry for mistake.

Round 2

Reviewer 1 Report

Previous comments were quite properly addressed. However, I suggest further minor changes to improve the quality of the manuscript:

Table 1 should be more synthetic, not just repeating what is written in the text. Therefore, I suggest reporting in the second column (named 'description' or 'composition') just the molar ratio of the components (using only their acronyms). For example, for sample AM0, AA:AMPS=4:1; for sample AM1, AA:AMPS:AN=4:1:4; for sample AM2, AA:AMPS:AN=4:1:8, and so on....

Paragraph 3.3 should be named 'antimicrobial activity'

Figure 1: could you please change the thickness of the graph lines? They are too thick, also compared to Fig. 2. In general, all the Figures/graphs do not seem to have a similar style and would be better to have them homogeneous.

Author contribution paragraph: 'software, X.X.' should be removed.

Author Response

Table 1 should be more synthetic, not just repeating what is written in the text. Therefore, I suggest reporting in the second column (named 'description' or 'composition') just the molar ratio of the components (using only their acronyms). For example, for sample AM0, AA:AMPS=4:1; for sample AM1, AA:AMPS:AN=4:1:4; for sample AM2, AA:AMPS:AN=4:1:8, and so on....

Answer: Table 1 modified

Paragraph 3.3 should be named 'antimicrobial activity'

Answer: It is corrected

Figure 1: could you please change the thickness of the graph lines? They are too thick, also compared to Fig. 2. In general, all the Figures/graphs do not seem to have a similar style and would be better to have them homogeneous.

Answer: Figure 2 modified

Author contribution paragraph: 'software, X.X.' should be removed.

Answer: it is deleted

Reviewer 2 Report

The manuscript has not improved according to reviewer comments. It looks almost similar with the previous submission. I can not agree with its publication in current form. Authors gave just simple responses as "it was done," and not showing what was done in your responses. Please note, the response could not be shorter than the question. I recommend to copy the changes made in the text and paste it in response to the reviewer´s report. 

Author Response

The paper reports on the antibacterial and anti-fungal properties of some synthetic-based cross-linked terpolymers. I consider that the manuscript is not suitable in the present form for publication in Materials journal because this paper has serious drawbacks, as it is presented in detail below:

  1. English needs improvement through out the manuscript and spelling should be checked thoroughly. There are many spelling mistakes and some phrases are not clear in the original paper. For example, in the introduction “Conventional techniques of polymerization such as the condensation and the free-radical polymerizations.”; “They are used mainly for the chemically cross-linked polymers”; “This variety of the kinds of monomers permits…” are not clear and should be rewritten.

Answer: The English was revised and new corrections were marker with red colour. The sentence was changed to be

Conventional techniques of polymerization such as the condensation and the free-radical polymerizations were used mainly for preparation of chemically cross-linked polymers [6]. These chemical crosslinking techniques produce a degradable or a non-degradable cross-linked polymer relying on the monomers types (ionic or noni-ionic) and nature of chemical bonds [7]. 

  1. The application of synthetic cross-linked terpolymer in life have abundantly increased through the recent years, due to their unique characteristics including mechanical strength, longer service life, the ability to absorb large amounts of water and being biocompatible” should be sustained by references.

 Answer: reference 10 added and the sentences clarified as

The using more than one monomer rather than crosslinker were used to obtain copolymers or terpolymers when two or three monomers were used, respectively [10]. This variety of the monomers permits the production of cross-linked polymers with desired properties utilized in different applications [10]. Typically, this technique requires an addition of a small amount of a cross-linking agent during the polymer production [11]. The application of synthetic cross-linked terpolymer in life have abundantly increased through the recent years, due to their unique characteristics including mechanical strength, longer service life, the ability to absorb large amounts of water and being biocompatible [10].

  1. The cross-linked terpolymers reported in this paper are based on synthetic polymers; therefore the discussion on natural polymers included in the introduction part is not related to the current study and should be removed. I recommend the addition of a similar discussion on for synthetic polymers already reported in literature.

 Answer: The biopolymers were widely used due to their natural availability beside their bioactivity. The synthetic crosslinked polymers require attention to select the desired monomers having lower toxicity and higher biological activity. The researches regarding application of synthetic hydrogels are few and we inserted only references reported the synthetic monomers such as

Tomar, R.S.; Gupta, I.; Singhal, R.; Nagpal, A. Synthesis of poly (acrylamide-co-acrylic acid) based superabsorbent hydrogels: Study of network parameters and swelling behaviour. Polymer-Plastics Technology and Engineering 2007, 46, 481-488.

  1. Uygun, M.; Kahveci, M.U.; Odaci, D.; Timur, S.; Yagci, Y. Antibacterial acrylamide hydrogels containing silver nanoparticles by simultaneous photoinduced free radical polymerization and electron transfer processes. Macromolecular Chemistry and Physics 2009, 210, 1867-1875.
  1. Lines 60-61: “anti-biological applications of cross-linked polymers” need clarifications and references.

 Answer: the anti-biological applications were clarified as anti-microbial, anti-fungail and antibacterial were clarified with references from 12 to 19

  1. In the Introduction, the novelty of the work is weakly emphasized. The synthesis of the acrylonitrile, acrylamide and 2-Acrylamido-2- methylpropane sulphonic acid cross-linked terpolymers has been already reported, refs. 20-22. In this context, the authors’ motivation for selecting this system needs stronger explanations. Why this system is better than others?

Answer: we inserted new sentences to clarify the reason of monomers selection as:

In the present work, different terpolymers were synthesized based on monomers, acrylonitrile (AN), acrylic acid (AA), acrylamide (Am), AMPS, benzoyl peroxide (BP) as an initiator and ethylene glycol dimethacrylate (EGDMA) cross-linker. These monomers were selected on the basis of their reactivity ratios to obtain neutral, and ionic terpolymers. Moreover, EGDMA was selected as reactive crosslinker to control the crosslinking densities of the prepared terpolymers.

  1. The Results and discussion section should be revised because some parts of it are ambiguous, difficult to follow, and don’t emphasize the value of this work. A clear, fluent and more critical interpretation of the results regarding the antibacterial and anti-fungal properties should be provided.

Answer: It is important to identify the chemical structure and thermal stability of the prepared terpolymers and the swelling measurements to show their interaction with water without dissolution and degradation. The antibacterial and antifungial part ws modified.

  1. In the FTIR spectra, Figure 1, please label the discussed peaks included in the main text. The optical pictures, Figure 4, have very low resolution and do not show any swelled gels.

 Answer: FTIR spectra were marked with the main functional groups as represented in modified figures 1 and 2

  1. In Table 1 and Table 2, please correct the name of the fungal bacterium as “Candida” and not “Candia”.

Answer: They were corrected.

  1. Where these cross-linked terpolymers are going to be used in real life? What is the relationship between antifungal properties and oil recovery? The last sentence included in the Conclusion part is not supported by the results presented in this paper.

Answer: the introduction section was modified to report that the importance of using hydrogels having antimicrobial activity to apply for drug delivery in antimicrobial areas. Moreover the discussion section improved to investigate the antimicrobial and anti-fungi activities of excellent swelled hydrogels

Round 3

Reviewer 2 Report

The manuscript could be accepted in the present form. 

Author Response

thanks